# The Role of the Functional Head Impulse Test with and without Optokinetic Stimuli in Vestibular Migraine and Acute Unilateral Vestibulopathy: Discovering a Dynamic Visual Dependence

**DOI:** 10.3390/jcm10173787

**Published:** 2021-08-25

**Authors:** Augusto Pietro Casani, Francesco Lazzerini, Ottavia Marconi, Nicola Vernassa

**Affiliations:** ENT Section, Medical, Molecular and Critical Area, Department of Surgical Pathology, Pisa University Hospital, 56122 Pisa, Italy; francilazzerini@gmail.com (F.L.); ottavavia@gmail.com (O.M.); n.vernassa@ao-pisa.toscana.it (N.V.)

**Keywords:** vestibular migraine, acute unilateral vestibulopathy, vertigo, visual dependence, visuo-vestibular interactions, functional head impulse test

## Abstract

(1) Background: Visually induced vertigo (i.e., vertigo provoked by moving visual scenes) can be considered a noticeable feature of vestibular migraines (VM) and can be present in patients suffering from acute unilateral vestibulopathy (AUV). Hypersensitivity to moving or conflicting visual stimulation is named visual dependence. (2) Methods: Visuo-vestibular interactions were analyzed via the functional Head Impulse Test (fHIT) with and without optokinetic stimulation (o-fHIT) in 25 patients with VM, in 20 subjects affected by AUV, and in 20 healthy subjects. We calculated the percentage of correct answers (%CA) without and with the addition of the optokinetic background (OB). (3) In VM groups, the %CA on the fHIT was 92.07% without OB and 73.66% with OB. A significant difference was found between %CA on the deficit side and that on the normal side in AUV, both without OB and with OB. (4) Conclusions: The fHIT results in terms of %CA with and without OB could be useful to identify the presence of a dynamic visual dependence, especially in patients suffering from VM. The difference in %CA with and without OB could provide instrumental support to help correctly identify subjects suffering from VM. We propose the use of the fHIT in clinical practice whenever there is a need to highlight a condition of dynamic visual dependence.

## 1. Introduction

Vertigo and dizziness are common complaints requiring accurate clinical and instrumental assessment. In otoneurological practice, acute unilateral vestibulopathy (AUV) and vestibular migraine (VM) represent two broad categories of vestibular disorders. AUV is the most important cause of sustained vertigo, characterized by a sudden onset of long-lasting vertigo with nausea and vomiting, gait instability, and a falling tendency toward the affected side without associated cochlear or central nervous system symptoms and signs [1]. VM is now considered the most common cause of episodic vestibular syndrome: the 1-year prevalence was estimated at 2.7% in a population-based study [2], a much higher number than previously reported [3]. Patients suffering from these two conditions can, at different stages of the disease, present some degree of visuo-vestibular symptoms: AUV patients can develop chronic symptoms such as head movement and visually induced dizziness, especially during head turns [4]. Patients with VM can manifest, along with various other types of vertigo (spontaneous vertigo, positional vertigo), visually induced vertigo, head-motion-induced vertigo, and head-motion-induced dizziness with nausea [5,6]. Visually induced vertigo (i.e., vertigo provoked by moving visual scenes, such as traffic or movies) can be considered a noticeable feature of VM and may persist between attacks [7]. Hypersensitivity to moving or conflicting visual stimulations is named “visual vertigo” or “visual dependence” (VD). A subject with VD privileges the degree of visual reliance during the central integration of sensory cues, and this condition reduces the ability to disregard visual clues in complex or conflicting visual environments [8]. The exact reason for this over-reliance on visual stimuli is still a matter of debate [9,10]; however, both AUV (probably because of an increase in the visual contribution in the sensory integration process) and VM (wherein a reduced capability to integrate different sensory signals reporting motion has been observed) [11,12,13] could facilitate the onset of symptoms compatible with VD [10].

To assess and quantify the role of VD in different physiological and pathological conditions, the rod and frame and rod and disc techniques [4,11,14], as well as the measurement of postural sway (induced by visual-roll-motion stimulation) by static posturography [4], have been used. Recently, a new test offered the possibility to obtain precise information about the patient’s ability to achieve clear vision of a target during rapid head turns. This test was named the functional head impulse test (fHIT), and it uses a dynamic paradigm (not a static one as in the rod and frame or rod and disc test) allowing us to obtain objective data about gaze stabilization during rapid passive head movements with the aim of assessing the functional performance of the vestibulo-ocular reflex (VOR). The f-HIT can be performed while optokinetic stimulation is given on the screen: this paradigm could detect the role of the visual component in maintaining balance by using a moving background while the test is administered [15,16]. A comparison of the responses without and with a moving background (optokinetic stimulation, o-fHIT) could give an indication of the effect on the ability to respond, expected to be much poorer in the case of a discrepancy between the perceived visual stimulus and vestibule-proprioceptive stimulus [13,15].

The aim of this study was to evaluate the potential visuo-vestibular interactions in two groups of patients, one suffering from AUV and the other from VM, by analyzing the results of the f-HIT both without and with optokinetic stimulation.

## 2. Materials and Methods

Between December 2018 and June 2020, we consecutively recruited 25 patients fulfilling the diagnostic criteria for VM [5]. All the VM patients were evaluated in the inter-critical phase and in the absence of any actual pharmacological prophylactic treatment. All patients also underwent a complete vestibular evaluation, which involves bedside examination, caloric reflex test, the video head impulse test (vHIT), and the functional head impulse test (fHIT). In the same period, we recruited 20 patients suffering from AUV. We diagnosed AUV as a syndrome characterized by rapid onset of severe dizziness without neurologic or audiologic symptoms, unidirectional horizontal nystagmus, unilateral vestibular areflexia/hyporeflexia on the bithermal caloric test, and a positive head impulse test result in the direction opposite to the fast phase of the nystagmus. Each patient with AUV underwent vestibular instrumental examination (the bithermal caloric test and video head impulse test, vHIT) after the disappearance of the acute symptomatology within 8–14 days after the onset of symptoms. The vHIT was performed by employing a dedicated device (ICS Impulse System; GN Otometrics): the subject was asked to stare at an Earth-fixed target (3 cm diameter spot located 1.5 m in front); then, 20 low-amplitude (10°–20°) but high-velocity head impulses (150–200 deg/s) were randomly administered on each side for every semicircular canal. The device software automatically calculates the average high-velocity VOR gain. The software also calculates the asymmetry index (AI%, normal values within 15% (% confidence limit 0–15%); this value resulted from our own data collected on a group of 50 normal subjects with age ranging from 20 to 80) between the right and left sides. The AI was calculated as:$$AI=\left[\frac{G_{L}-G_{R}}{G_{L}+G_{R}}\right]\times 100$$
where *G_R_* denotes the right-sided mean gain and *G_L_* denotes the left-sided mean gain. The AI% shows differences between the sides in terms of high-velocity VOR gain [17]. All patients underwent brain magnetic resonance imaging to rule out central nervous system involvement. Patients suffering from Ménière’s disease, sudden sensorineural hearing loss, or recent head trauma were excluded. Altogether we considered a control group consisting of 20 healthy subjects, excluding subjects who suffered in the past from AUV who reported recent episodes of benign paroxysmal positional vertigo or a personal or familial history of migraine. All the subjects underwent the fHIT using the Beon Solution System^®^ (Beon Solution srl, Treviso, Italy): the subject was seated 1.5 m away from the computer screen (Full HD, resolution 1920 × 1080, refresh rate 60 Hz) wearing an accelerometer firmly tightened on the forehead by a band. Static visual acuity was measured at the beginning of the fHIT using a Landolt C optotype; the dimensions of the visual stimulus were normalized according to the characteristics of the subject examined. Subsequently, the examiner quickly and randomly administered head thrusts on the horizontal plane (at least 10 turns in each direction), scanning a wide range of head angular accelerations (between 2000 and 6000 °/s^2^). During the movement, a symbol was displayed on the monitor for 80 milliseconds [18]. The subject was asked to recognize the symbol that appeared on the monitor during the head movement and to press the equivalent key. The operator performed a minimum of 15 head impulses in each direction on the plane of the horizontal semicircular canal. The parameter taken into consideration was the average percentage of correct answers (%CA) with respect to the total number of answers presented for each frequency band examined (acceleration bins). For statistical analysis, only head accelerations above 3000 °/s^2^ were considered. A value of more than 80% of CA was considered normal [16,18]. The test was then repeated with the addition of optokinetic stimuli (optokinetic background, OB), that is, a moving background constituted by a rotating cloud of yellow dots, named optokinetic fHIT (o-fHIT), displayed on the screen while the above-described test was performed. To evaluate the results of the test, the following procedure was carried out:The answers for the head rotation to the right and those for the rotation to the left were separated and considered different tests; in this way, the number of exams was double the number of subjects;For each test, the difference in %CA without and with the OB was evaluated.

All the subjects were submitted to the Italian version of the situational vertigo questionnaire (SVQ) [19] to identify whether vertigo symptoms are provoked or exacerbated by a specific disorienting visual context. The total score was then calculated as the sum of the single item scores divided by 19 minus the number of never experienced situations (total score/19– (number of “never experienced” answers)). 

Ethical review and approval by the local Institutional Board (Comitato Etico Azienda Ospedaliero-Universitaria Pisana, Pisa, Italy) were waived for this study. Due to its retrospective nature, it was not set up as part of a research project. Furthermore, the study does not include new experimental diagnostic protocols, and the patients included in the study were diagnosed according to national guidelines. Written informed consent was obtained from all participants, and the study was conducted in accordance with the 1964 Declaration of Helsinki.

### Statistical Study

The data obtained were statistically compared using the variance analysis method. The values obtained were distributed in a normal way; therefore, it was possible to perform the following:(1)A study of variance (one-way ANOVA), which allowed us to reject the null hypothesis of homogeneity of the groups (*p* < 0.05);(2)A statistical comparison between patients with migraine vertigo (one-way *t*-test) and the normal group;(3)A statistical comparison between patients with compensated acute vestibular deficit and the normal group (one-way *t*-test);(4)A comparison between patients suffering from migraine and those suffering from compensated acute vestibular deficit (one-way *t*-test).

A study of variance with Tukey’s correction was also carried out; this was also significant due to the rejection of the null hypothesis between the groups (*p* < 0.05).

## 3. Results

The demographic data, clinical features, and distribution of %CA for the three evaluated groups are reported in Table 1 (patients with VM), Table 2 (patients with AUV), and Table 3 (control group, healthy subjects). The mean age of the AUV group (59.6 years, ranging from 39 to 78, 11 males and 9 females) was slightly higher than that of the VM group (53.4 years, ranging from 32 to 67, 20 females and 5 males). The control group comprised 20 healthy subjects (10 males and 10 females, age ranging from 39 to 70, mean 53.8). Regarding the patients suffering from VM, the duration of symptoms was 9.8 ± 6.5 months (range 6–31). All patients reported episodic attacks of rotational vertigo with no symptoms in the inter-critical phase; nine subjects suffered from chronic unsteadiness between attacks. Sixteen patients reported head motion intolerance and twenty suffered from motion sickness. Regarding the instrumental examination, we found a caloric hyperreflexia in eight patients. Five patients showed a positional persistent nystagmus, associated in one case with positivity of the head shaking test.

All patients suffering from AUV, according with the initial diagnosis, showed a unilateral caloric weakness and a pathological asymmetry index on the vHIT (11 cases of left AUV and 9 cases of right AUV, referred to in the text as AUV-L and AUV-R, respectively). Only two patients in the AUV group reported the presence of motion sickness.

In the VM group, the %CA on the fHIT was 90.48% on the left side and 93.67% on the right side; with the OB (optokinetic stimulation), the %CA was 73.69% on the left side and 73.63% on the right side (Figure 1). A significant percentage (58%) of patients with VM showed a significant decrease in %CA with the use of the OB as compared with normal subjects. Considering the %CA on the left and on the right simultaneously, in VM groups, the %CA on the fHIT was 92.07% without OB and 73.66% with OB. A statistically significant difference was found between the %CA values of the VM group with and without OB (*p* < 0.001), between the %CA values of the VM group without OB and the CTRL group without OB (*p* < 0.01), and, most of all, between the %CA values of the VM group with OB and the CTRL group with OB (*p* < 0.001). Meanwhile, no significant difference was found between the %CA values of the CTRL group with and without OB (Figure 1).

The %CA values in the AUV-R group were 78.12% on the left and 39.81% on the right without OB; with OB, the %CA values were 72.23% on the left and 32.97% on the right. In the AUV-L group, the %CA values without OB were 53.60% on the left and 85.30% on the right; with OB, the %CA values were 40.25% on the left and 71.28% on the right. In the control group, the %CA values were 97.75% on the left and 97.5% on the right without OB and 94.71% on the left and 96.19% on the right with OB. No significant difference was found between the left and right %CA values, without or with OB, between the VM and CTRL groups (Figure 2). When analyzing the change in %CA with the optokinetic background in the AUV group, we considered AUV-R and AUV-L together as a single group and analyzed the %CA on the deficit side, whether left or right, and the %CA on the normal side, whether left or right. A significant difference was found between the %CA values on the deficit side and the normal side in the AUV-L and AUV-R groups, both without OB (*p* < 0.001 and *p* < 0.001, respectively) and with OB (*p* < 0.001 and *p* < 0.001, respectively). A significant difference was also found in the %CA values for AUV-R and AUV-L on the normal side and on the deficit side and the %CA values in the CTRL group without OB (*p* < 0.05, *p* < 0.05 and *p* < 0.001, *p* < 0.001, respectively) and with OB (*p* < 0.01, *p* < 0.01 and *p* < 0.001, *p* < 0.001, respectively) (Figure 2). The data relating to the tests between groups confirmed the results of the analysis of variance, allowing us to reject the null hypothesis that the subjects of the study are from the same population, but also allowing us to say that the populations of migraineurs and those with vestibular deficits are distinct among them (*p* < 0.001).

The mean difference in the errors with and without the optokinetic background (∆OB) in the VM group was 18.36. A total of 58% of patients with VM showed a significant decrease in the %CA with the OB, considered as a %CA decrease greater than 2SD of the means of the %CA. In the AUV group, this percentage was 36%, meanwhile in the control group that was limited to 7%. ∆OB value in the CTRL group was 2.15. Finally, ∆OB in the AUV group was 10.4 (Figure 3). A statistically significant difference was found between ∆OB values in the VM and CTRL groups (*p* < 0.001) and ∆OB values in the AUV and CTRL groups (*p* < 0.001). Further, a less significant difference was found between ∆OB values in the VM and AUV groups (*p* < 0.05) (Figure 3).

Regarding the reported SVQ scores, the mean score for the VM group was 40.36 (ranging from 21.0 to 61.0); in the AUV group, the mean score was 38.35 (from 28.0 to 53.0); while in the control group, the mean score was 5.25 (from 0.0 to 12.0). The difference between groups was statistically significant (*p* < 0.001). A significant difference was found between the SVQ scores for the VM and AUV patients and those for the control group (*p* < 0.001 and *p* < 0.001, respectively). On the other hand, no significant difference was found between the VM and AUV groups regarding SVQ scores (*p* = 0.530) (Figure 4). The SVQ results were significantly correlated with groups (Spearman’s coefficient = 0.657, *p* < 0.001) and significantly negatively correlated with %CA with and without OB (Pearson’s coefficient = −0.677, *p* < 0.001 and Pearson’s coefficient = −0.472, *p* < 0.001, respectively).

## 4. Discussion

Achieving an adequate spatial orientation requires the integration of vestibular, proprioceptive, and visual inputs. Some normal subjects show an overreliance on visual stimuli, probably due to a lack of confidence in vestibular or somatosensory input [8]. This condition is termed VD and could be considered a perceptual trait or cognitive style variably expressed in the general population [20,21]; more recently, it was named by the Barany Society as visual-induced dizziness [22]. In patients suffering from vestibular disease, the interactions between visual and vestibular input could be impaired [4,11,23] and for this reason a complex visual input could induce a postural destabilization [24]. In this situation, the inability to ignore a moving and/or complex visual scene (optokinetic stimuli) can lead to it being confused with self-motion, resulting in instability. Following AUV, VD could be a strong predictor of poor clinical outcome [9]; patients with VM showed larger errors in upright perception during static head tilts compared with normal subjects [25] and a significant reduction in motion thresholds [26]. Further studies show evidence suggesting an increased visual dependence, as measured with an optokinetic rotating disc [11,27,28,29]. All these experiences indicate that patients with VM could develop some kind of dysfunction in neural mechanisms that subserve spatial orientation, probably related to a difficulty in integrating sensory information that encodes the positions of the eye, head, and body in order to maintain perception of the upright and correct gaze stabilization during complex visual scenes.

Recently a new method to evaluate the functional performance of the vestibulo-ocular reflex (VOR) during passive head movement was proposed: the fHIT quantifies the percentage of correctly recognized optotypes (using a relatively large, fixed-size optotype) during head impulses scanning a wide range of head angular accelerations [18]. The fHIT identifies the extent to which the actual stabilization ability is impaired as head acceleration increases [15]; by combining this test with optokinetic stimulation (a rotating background or confounding screen), impaired integration between visual and vestibular stimuli can be detected [13].

Our results indicated that a significant percentage (58%) of patients with VM showed a significant decrease in the %CA when using a confounding screen (optokinetic background, o-fHIT) as compared with normal subjects. This behavior was also detected in AUV patients, but they showed a clear difference according to the affected side: A reduction in %CA passing from a normal to a OB was determined by the results on the affected side only. In other words, the deterioration of the performance on the fHIT passing from a normal to a optokinetic background is not side-dependent in VM, while in AUV patients, the reduction in the %CA when using optokinetic stimulation is detected mostly when rotating the head towards the affected side. In these patients, the abnormalities in the results of the o-fHIT on the affected side are clearly related to altered gaze control because of a reduction in the VOR gain. On the contrary, the impairment of multisensory integration showed by VM patients during the fHIT with optic flow stimulation could be considered a consequence of visuo-vestibular conflict related to central nervous system abnormalities, frequently encountered in these kinds of diseases [30,31]. Versino et al. [13] reported functional vestibular impairment using the fHIT, especially when the head impulse test was performed in combination with optokinetic stimulation. Our results seem to confirm this observation: our group of AUV patients showed, as expected, a reduced capability to correctly recognize a visual target during the fHIT when the head was turned to the affected side. This impairment slightly increased when an integration of visuo-vestibular input was necessary, as happens when an optokinetic stimulus is presented, inducing a decrease in the %CA. In VM patients, our results indicate that the %CA was normal in most of the subjects, and only when using the optokinetic stimulation did this parameter strongly decrease (with no difference between the two sides) not only with respect to the control group but also in comparison with the AUV group, showing that visual dependence is the cause of the decay in the responses when the background is in motion. Recently, Al-Sharif et al. [32] studied the effects of visual–vestibular mismatch using a specific questionnaire and a computerized rod and frame test; they reported that the presence of VD was significantly higher in patients suffering from headache and dizziness than in subjects with headache only. Visual dependence was measured as the error to a subjective visual vertical using a computerized rod and frame test. Despite the different technique used (the fHIT assesses the functional performance of the VOR during passive head impulses, while the rod and frame test assesses verticality perception in static conditions), our results are in line with this observation, indicating the very important involvement of the vestibule in VM causing difficulty in resolving conflicting visuo-vestibular signals in dynamic conditions (head impulses), as demonstrated by the significant differences in the delta OB values between our three groups of subjects. Another result of our study is represented by the high percentage (%) of patients with VM suffering from motion sickness with respect to the AUV patients or controls. It is probable that motion sickness and reduced capability to integrate visual vestibular signals (as happens in the o-fHIT) have a common denominator: a sensory conflict could modify the awaited experiences, causing a discordance between the expected and the real perception, leading to motion sickness or visual vertigo or dizziness [33,34].

In our study, the evaluation of the subjects’ self-feeling was conducted using the Italian version of the SVQ [19]. This questionnaire was introduced by Jacob et al. in 1993 to quantify space and motion discomfort in patients suffering from anxiety and/or balance disorders and subsequently adapted to evaluate the presence of visual vertigo [4]. Our results regarding the SVQ indicate that visual-induced vertigo and dizziness are very common in both VM and AUV patients. This result is not surprising because migrainous patients (with and without associated vertigo) are highly susceptible to increased visuo-vestibular conflict [19,35,36]. High SVQ levels were described in patients with vestibular disease [4,8,11,37], and our results seem to confirm this observation. Our group of AUV patients was evaluated a few days after the onset of the disease: this condition is probably the cause of the higher SVQ score encountered in our cohort. Surely the presence of increased SVQ levels and low %CA with optokinetic stimulation could indicate the possibility of an evolution towards chronic symptoms after AUV and could identify the need for early rehabilitation treatment using a desensitization technique [9,38], but more data should be obtained at a more advanced stage of the disease. In other words, our results indicate that symptoms of visual vertigo or impaired visuo-vestibular interactions are very frequent in the subacute phase of AUV with a similar rate to that encountered in some patients suffering from VM. In a high percentage of patients suffering from AUV, the process of vestibular compensation could induce a reduction in this kind of symptom, but its persistence is significantly higher in those with worse clinical outcomes [9].

Another important topic is the correlation between the presence of psychopathological conditions (anxiety/depression) and impaired visuo-vestibular interactions. Previous studies reported conflicting results [4,39], but recently Teggi et al. [40] studied 25 patients diagnosed as affected by PPPD and showed that the fHIT with optokinetic stimulation provoked more error readings in these patients than in controls, supporting the hypothesis that increased anxiety could lead to an alteration of visuo-vestibular interactions. This observation could introduce a bias in our study, considering that we did not evaluate the levels of anxiety/depression in our cohort of subjects. For this reason, we consider this aspect a study limitation. However, the presence of some psychopathological aspects in VM is very common, so much so that the combination of a balance disorder, migraine, and anxiety has been named migraine–anxiety-related dizziness [41] based on shared mechanisms in afferent interoceptive information and central nervous system processing [42]. Another study limitation is represented by characteristics of the study design: a prospective study with a large cohort of patients would surely bring a better assessment of the role of the fHIT in diagnosing visual dependence in subjects suffering from VM and AUV and in evaluating the efficacy of rehabilitative and/or pharmacological treatment. Finally, we do not have data about a possible difference in the VM patients between fHIT results during vertigo attacks and those between attacks.

## 5. Conclusions

The results of the fHIT in terms of %CA with and without OB could be very useful to identify the presence of dynamic visual dependence, especially in patients suffering from VM. This observation has a twofold clinical impact: Firstly, the patients with a clear reduction in %CA with CS could benefit from a rehabilitation treatment based on a visual motion desensitization technique [38]. Secondly, taking into consideration the fact that VM is currently diagnosed only using clinical criteria, the availability of a vestibular instrumental hallmark (such as the difference in %CA with and without OB) could provide some support to correctly identify this category of patients, particularly when the international diagnostic criteria for VM are not completely met. In this case, a negative or inconclusive clinical and instrumental examination associated with an altered fHIT only under a optokinetic background could support the diagnosis of vestibular migraine. Similarly, in acute vestibular deficit, the fHIT carried out with and without a OB can reveal a visual dependence. We propose the use of the fHIT in clinical practice whenever there is a need to highlight a condition of dynamic visual dependence in order to undertake the correct therapeutic strategy.

## Figures and Tables

**Figure 1 jcm-10-03787-f001:**
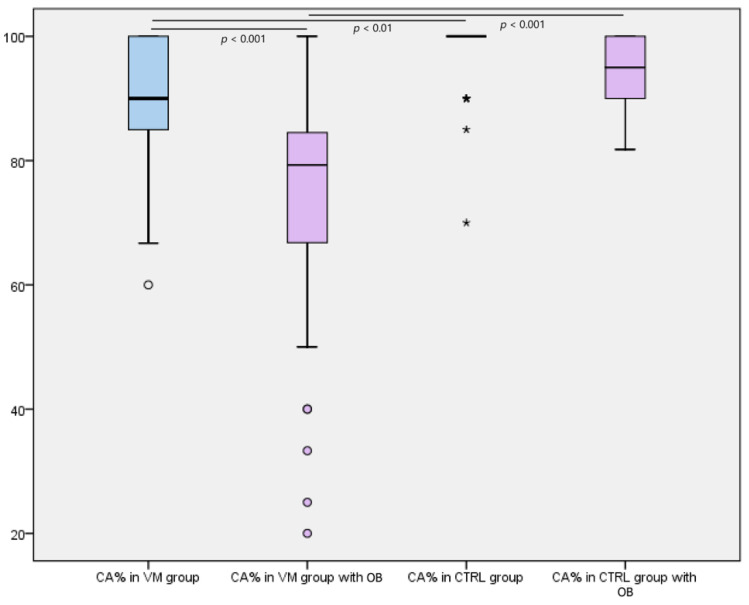
Boxplot showing the percentages of correct answers on the fHIT in the VM group and the CTRL group without and with optokinetic background (OB). A statistically significant difference was found between the %CA values of the VM group with and without OB (*p* < 0.001), between the %CA values of the VM group without OB and the CTRL group without OB (*p* < 0.01), and, most of all, between the %CA values of the VM group with OB and the CTRL group with OB (*p* < 0.001). No significant difference was found between the %CA values of the CTRL group with and without OB. * = Extreme values outliers, considering a step of 1.5 × IQR (Interquartile range).

**Figure 2 jcm-10-03787-f002:**
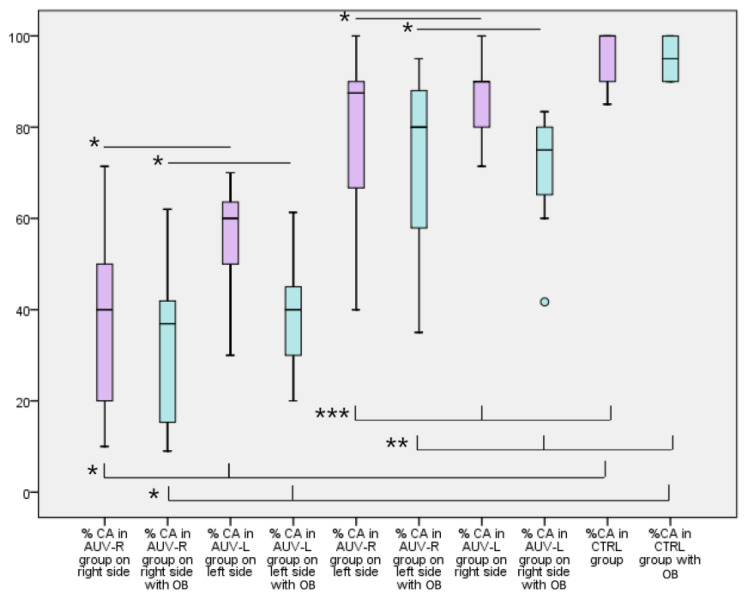
Boxplot showing the percentages of correct answers in the AUV group on the right and left sides without and with the optokinetic background (OB) and the %CA in the control group without and with OB. A significant difference was found between %CA values on the deficit side and the normal side in the AUV-L and AUV-R groups, both without OB (*p* < 0.001 and *p* < 0.001 [*], respectively) and with OB (*p* < 0.001 and *p* < 0.001 [*], respectively). A statistically significant difference was also found in the %CA values in AUV-R and AUV-L on the normal side and on the deficit side and the %CA values in the CTRL group without OB (*p* < 0.05, *p* < 0.05 [***] and *p* < 0.001, *p* < 0.001 [*], respectively) and with OB (*p*< 0.01, *p* < 0.01 [**] and *p* < 0.001, *p* < 0.001 [*], respectively).

**Figure 3 jcm-10-03787-f003:**
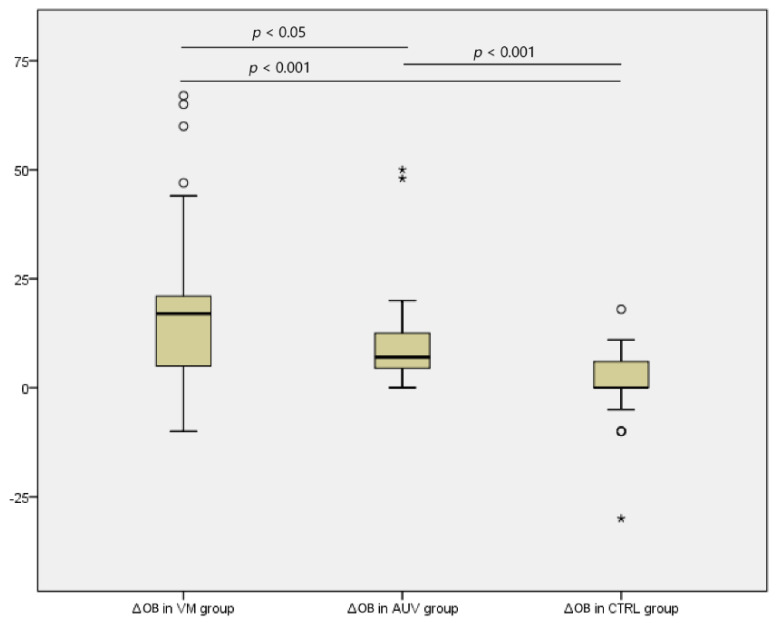
Boxplot showing the ∆OB values (the absolute value of the difference in the errors with and without optokinetic background, OB) in the VM, AUV, and CTRL groups. A statistically significant difference was found between the ∆OB values in the VM and CTRL groups (*p* < 0.001), the ∆OB values in the AUV and CTRL groups (*p* < 0.001), and the ∆OB values in the VM and AUV groups (*p* < 0.05). * = Extreme values outliers, considering a step of 1.5 × IQR (Interquartile range).

**Figure 4 jcm-10-03787-f004:**
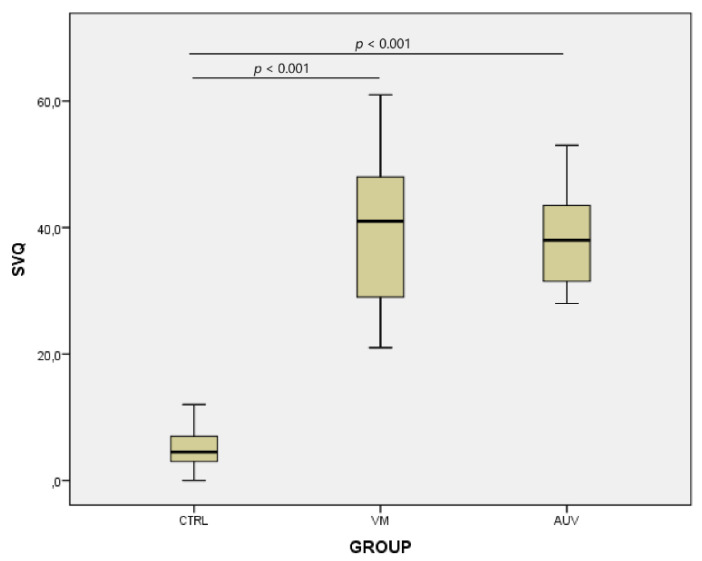
Boxplot showing the SVQ scores in the VM, AUV, and CTRL groups. Significant differences were found between the SVQ scores of VM and the AUV patients and those of the control group (*p* < 0.001 and *p* < 0.001, respectively). No significant differences were found between the VM and AUV groups regarding SVQ scores.

**Table 1 jcm-10-03787-t001:** The vestibular migraine group (VM): demographic data, clinical features, distribution of percentage of correct answers (%CA) on the left and right sides without and with the optokinetic background (OB), and situational vertigo questionnaire (SVQ) score. EV, episodic vertigo; Dizz, dizziness; MS, motion sickness; HMI, head motion intolerance; Hy, hyperreflexia; PosNy, positional persistent nystagmus; HST+, positive head shaking test.

Sex, Age	%CA on the Left	%CA on the Right	%CA on the Left + OB	%CA on the Right + OB	SVQ	Instrumental Signs	Symptoms
F, 41	100	85	80	100	28	PosNy; Hy	EV, Dizz, MS, HMI
F, 59	80	100	90	90	21	none	EV, MS
M, 43	95	90	70	94.4	32	Hy	EV, MS
F, 61	100	100	90	100	25	none	EV
F, 44	100	90	100	55.6	45	Hy	EV, MS, HMI
F, 63	100	100	65	76	34	PosNy	EV, Dizz, MS, HMI
F, 41	90	100	70	100	28	none	EV, Dizz, MS, HMI
F, 46	100	100	45	77.8	45	Hy	EV, MS, HMI
F, 69	85	80	40	20	56	Hy	EV, MS, HMI
F, 42	90	100	90	33.8	61	none	EV, Dizz, MS, HMI
M, 32	80	66.7	90	40	48	none	EV, MS, HMI
F, 44	100	90	75	90	32	PosNy	EV, Dizz
F, 55	90	100	100	81.8	29	none	EV, Dizz, MS, HMI
F, 51	75	100	100	72.7	45	none	EV, MS, HMI
M, 45	76.9	70	72	55.6	41	PosNy; HST+	EV, Dizz
M, 38	100	100	65	100	25	none	EV, MS, HMI
F, 62	80	100	65	82.4	39	Hy	EV, Dizz, MS, HMI
F, 47	100	90	80	22.2	51	none	EV, MS, HMI
F, 39	100	100	60	85	34	none	EV, Dizz
F, 38	90	100	80	66.7	52	Hy	EV, MS, HMI
F, 43	95	90	90	70	48	none	EV, MS
F, 57	70	100	50	80	46	Hy	EV, MS, HMI
M, 43	90	100	85	90	28	none	EV, MS
F, 68	90	100	25	70	60	Pos Ny	EV
F, 41	85	90	65	86.7	56	none	EV, MS, HMI
**Mean**	**90.47**	**93.66**	**73.68**	**73.62**			

**Table 2 jcm-10-03787-t002:** The acute unilateral vestibulopathy group (AUV): demographic data, distribution of percentage of correct answers (%CA) on the left (L) and right (R) sides without and with the optokinetic background (OB), situational vertigo questionnaire (SVQ) score, and asymmetry index (AI) detectable via the vHIT.

Sex, Age	%CA on the Left	%CA on the Right	%CA on the Left + OB	%CA on the Right + OB	SVQ	Instrumental Signs	Symptoms
F, 41	100	85	80	100	28	PosNy; Hy	EV, Dizz, MS, HMI
F, 59	80	100	90	90	21	none	EV, MS
M, 43	95	90	70	94.4	32	Hy	EV, MS
F, 61	100	100	90	100	25	none	EV
F, 44	100	90	100	55.6	45	Hy	EV, MS, HMI
F, 63	100	100	65	76	34	PosNy	EV, Dizz, MS, HMI
F, 41	90	100	70	100	28	none	EV, Dizz, MS, HMI
F, 46	100	100	45	77.8	45	Hy	EV, MS, HMI
F, 69	85	80	40	20	56	Hy	EV, MS, HMI
F, 42	90	100	90	33.8	61	none	EV, Dizz, MS, HMI
M, 32	80	66.7	90	40	48	none	EV, MS, HMI
F, 44	100	90	75	90	32	PosNy	EV, Dizz
F, 55	90	100	100	81.8	29	none	EV, Dizz, MS, HMI
F, 51	75	100	100	72.7	45	none	EV, MS, HMI
M, 45	76,9	70	72	55.6	41	PosNy; HST+	EV, Dizz
M, 38	100	100	65	100	25	none	EV, MS, HMI
F, 62	80	100	65	82.4	39	Hy	EV, Dizz, MS, HMI
F, 47	100	90	80	22.2	51	none	EV, MS, HMI
F, 39	100	100	60	85	34	none	EV, Dizz
F, 38	90	100	80	66.7	52	Hy	EV, MS, HMI
F, 43	95	90	90	70	48	none	EV, MS
F, 57	70	100	50	80	46	Hy	EV, MS, HMI
M, 43	90	100	85	90	28	none	EV, MS
F, 68	90	100	25	70	60	Pos Ny	EV
F, 41	85	90	65	86.7	56	none	EV, MS, HMI
**Mean**	**90.47**	**93.66**	**73.68**	**73.62**			

**Table 3 jcm-10-03787-t003:** The control group (CTRL): demographic data, distribution of percentage of correct answers (%CA) on the left and right sides, without and with the optokinetic background (OB), and situational vertigo questionnaire (SVQ) score.

Sex, Age	%CA on the Left	%CA on the Right	%CA on the Left + OB	%CA on the Right + OB	SVQ
F, 45	100	100	100	100	0
M, 66	100	100	95	100	2
M, 48	100	100	95	95	3
F, 49	85	100	90	90	2
M, 70	90	100	100	90	5
M, 44	90	100	100	100	4
M, 68	100	100	90	100	7
M, 42	100	100	95	100	2
F, 50	100	90	90	100	4
M, 58	100	100	81.8	88.9	8
M, 63	100	100	90	90	7
F, 39	100	100	100	100	3
F, 66	100	90	95	90	10
F, 64	100	100	95	100	4
F, 54	100	100	100	95	7
F, 54	100	100	92.9	100	8
M, 65	100	100	94.5	95	7
F, 45	90	70	90	100	12
F, 39	100	100	100	90	6
M, 47	100	100	100	100	4
**Mean**	**97.75**	**97.5**	**94.71**	**96.20**	

## Data Availability

Data are contained in the article.

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
