# Peer review of "The Role of the Functional Head Impulse Test with and without Optokinetic Stimuli in Vestibular Migraine and Acute Unilateral Vestibulopathy: Discovering a Dynamic Visual Dependence"

_jcm, 2021, doi:10.3390/jcm10173787_

Round 1
Reviewer 1 Report
The manuscript presents a study on visual dependence in vestibular migraine (VM) and acute unilateral vestibulopathy (AUV) patients using the functional head impulse test (fHIT) without and with an optokinetic stimulation, while also measuring VOR gain and collecting SVQ answers.
They found that the change in the percentage of correct answers (%CA) with the addition of the optokinetic stimulus was significant both in VM and AUV patients, yet larger in VM patients. The group of VM patients, though, showed reduced %CA with the optokinetic stimulation on both head rotation directions, while the AUV group showed a significant decrease only on rotations towards the affected side.
The manuscript is scientifically sound and generally well written, yet a thorough spell-check is needed.
Specific comments
Figures 1, 2 and 3 labels refer to CS presumably instead of OB (optokinetic background).
Lines 222 and following report on the change of %CA with the addition of the OB for AUV group. Please explain how this value was computed with respect to the two groups AUV-R and AUV-L.
Line 275 The information that 58% of VM patients showed a significant decrease in %CA with the optokinetic stimulus should be reported in the results section. Similar information should also be reported in the results for the AUV group.
Line 92 the equation refers to GA while the text refers to AI. Please correct.
Line 94 'gain' should be added to the sentence '...high-velocity VOR'
Line 99 and following the two sentences should be merged as '... who did report recent episodes of benign paroxysmal positional vertigo or a personal or familiar history of migraine.'
Line 109 The fHIT actually displays the optotype for about 80 ms (see citation 13)
Many spelling and syntactic errors related to number agreement and verb tenses need a thorough English revision.
Author Response
Response to Reviewer 1 Comments
The manuscript presents a study on visual dependence in vestibular migraine (VM) and acute unilateral vestibulopathy (AUV) patients using the functional head impulse test (fHIT) without and with an optokinetic stimulation, while also measuring VOR gain and collecting SVQ answers.
They found that the change in the percentage of correct answers (%CA) with the addition of the optokinetic stimulus was significant both in VM and AUV patients, yet larger in VM patients. The group of VM patients, though, showed reduced %CA with the optokinetic stimulation on both head rotation directions, while the AUV group showed a significant decrease only on rotations towards the affected side.
The manuscript is scientifically sound and generally well written, yet a thorough spell-check is needed.
We thank the reviewer for his valuable comments, and we are very honored for this good general comment to our study
Specific comments
POINT 1: Figures 1, 2 and 3 labels refer to CS presumably instead of OB (optokinetic background).
Response 1: we are sorry for the mistake; in the first version of the manuscript, we adopted the term “confounding screen (CS)” to indicate a moving or conflicting visual stimulation. Successively we substituted CS with optokinetic background and during the formatting process the correction did not regard the figure labels. We made the requested correction
POINT 2: Lines 222 and following report on the change of %CA with the addition of the OB for AUV group. Please explain how this value was computed with respect to the two groups AUV-R and AUV-L.
Response 2: we thank the reviewer for allowing us to better explain the method used in comparing the AUV groups at line 206: when analysing the change in %CA with the confounding screen in the AUV group, we considered AUV-R and AUV-L as a single group and analysed the %CA on the deficit side, whether left or right, and the %CA on the normal side.
POINT 3: Line 275 The information that 58% of VM patients showed a significant decrease in %CA with the optokinetic stimulus should be reported in the results section. Similar information should also be reported in the results for the AUV group.
Response 3: I thank the reviewer for the possibility of a key clarification. At line 221, we provided a clarification of the point.
POINT 4: Line 92 the equation refers to GA while the text refers to AI. Please correct.
Response 4: thanking the auditor's careful observation, we have done the correction requested at line 94.
POINT 5: Line 94 'gain' should be added to the sentence '...high-velocity VOR'
Response 5: we apologise for the typing error; we have corrected by completing the sentence at line 96.
POINT 6: Line 99 and following the two sentences should be merged as '... who did report recent episodes of benign paroxysmal positional vertigo or a personal or familiar history of migraine.'
Response 6: we have made the proposed correction, which proved very useful in making the speech more fluid.
POINT 7: Line 109 The fHIT displays the optotype for about 80 ms (see citation 13)
Response 7: thanking the reviewer for the indication, we made the correction requested at line 109.
POINT 8: Many spelling and syntactic errors related to number agreement and verb tenses need a thorough English revision.
Response 8: as suggested the manuscript is underdoing to extensive English revisions using MDPI author service (enclosed you can find a English-editing certificate)
Reviewer 2 Report
The healthy control group is not mentioned in the abstract.
Healthy subjects should not be named "patients".
In VM, the terms "attack" and "between attacks" should be used.
It would be interesting to see whether there are differences between the attack and the time between attacks.
Author Response
Response to Reviewer 2 Comments
We thank the reviewer for his valuable comments, and we made some corrections according with the reviewer’s suggestions
POINT 1: The healthy control group is not mentioned in the abstract.
Result 1: As suggested by the reviewer, we add a sentence in the abstract regarding the presence of a control group.
POINT 2: Healthy subjects should not be named "patients".
Result 2: we are sorry for the mistake; we replaced “healthy subjects” instead of “patients”
POINT 3: In VM, the terms "attack" and "between attacks" should be used.
Result 3: we studied the VM patients in the intercritical phase of the disease (between attacks). We reported the symptomatology of VM patients, indicating that all the patients reported episodic vertigo with no symptoms between attacks. Only 9 VM patients reported chronic dizziness between attacks. We clarified this clinical behavior in the results section
POINT 4: It would be interesting to see whether there are differences between the attack and the time between attacks.
Result 4: we thank the reviewer for this interesting observation. Since the difficulties to evaluate VM patients during the attack, we have no data regarding the clinical and instrumental differences between the acute versus the intercritical phase in VM patients. However, we added a sentence in the limitation study section of our manuscript